# Delay and Energy-Efficiency-Balanced Task Offloading for Electric Internet of Things

Yong Wei [1], Huifeng Yang [1], Junqing Wang [1], Xi Chen [1], Jianqi Li [2], Sunxuan Zhang [2,*] and Biyao Huang [2]

[1] State Grid Hebei Electric Power Co., Ltd., Information and Communication Branch, Shijiazhuang 050021, China; weiy@he.sgcc.com.cn (Y.W.); yhf@he.sgcc.com.cn (H.Y.); xtgs_wangjq@he.sgcc.com.cn (J.W.); xtgs_chenx@he.sgcc.com.cn (X.C.)

[2] Global Energy Interconnection Research Institute Co., Ltd., Beijing 102209, China; lijianqi@geiri.sgcc.com.cn (J.L.); huang_dianqi@126.com (B.Y.)

* Correspondence: sunxuan_zhang@163.com

**Abstract:** With the development of the smart grid, massive electric Internet of Things (EIoT) devices are deployed to collect data and offload them to edge servers for processing. However, the task of offloading optimization still faces several challenges, such as the differentiated quality of service (QoS) requirements, decision coupling among multiple devices, and the impact of electromagnetic interference. In this paper, a low-complexity delay and energy-efficiency-balanced task offloading algorithm based on many-to-one two-sided matching is proposed for an EIoT. The proposed algorithm shows its novelty in the dynamic tradeoff between energy efficiency and delay as well as in low-complexity and stable task offloading. Specifically, we firstly formulate the minimization problem of weighted difference between delay and energy efficiency to realize the joint optimization of differentiated QoS requirements. Then, the joint optimization problem is transformed into a many-to-one two-sided matching problem. Through continuous iteration, a stable matching between devices and servers is established to cope with decision coupling among multiple devices. Finally, the effectiveness of the proposed algorithm is validated through simulations. Compared with two advanced algorithms, the weighted difference between the energy efficiency and delay of the proposed algorithm is increased by 29.01% and 45.65% when the number of devices is 120, and is increased by 11.57% and 22.25% when the number of gateways is 16.

**Keywords:** electric Internet of Things (EIoT); differentiated quality of service (QoS) requirements; task offloading; many-to-one two-sided matching

## 1. Introduction

The electric Internet of Things (EIoT) is a kind of industrial-level Internet of Things (IoT) [1,2] applied to the smart grid. By effectively integrating all kinds of communication and computing resources, the EIoT improves the information level of the electric power system and provides network support for the construction of the smart grid [3,4]. With the deployment of massive sensing devices, the data volume and computing requirements of the EIoT's perception of information have shown explosive growth [5]. The data processing of the EIoT has high requirements on energy efficiency and delay. On one hand, battery-powered EIoT devices have limited energy [6–8]. Improving energy efficiency can increase the utilization of energy resources and extend the network life cycle. On the other hand, power services such as differential protection and precise load control require low latency. However, traditional cloud computing where cloud servers locate far away from EIoT devices causes large transmission delay and low energy efficiency [9,10].

Edge computing provides computing services through an integrated open platform near the data source [11]. Its applications are initiated on the device side to reduce the transmission delay and energy consumption, thereby resulting in faster network service response and higher energy efficiency for data processing. By combining edge computing

with EIoT gateways, devices can offload data to one of the nearby gateways for processing to jointly improve delay and energy efficiency performance [12]. However, the task offloading optimization for an edge-computing-based EIoT still faces some challenges.

First of all, task offloading should be optimized based on dynamically varying network-state information to improve both delay and energy efficiency performance [13,14]. The challenge is that improving delay may deteriorate energy efficiency and vice versa. Therefore, a well-balanced tradeoff between delay and energy efficiency is required. Second, given the intensive deployment of EIoT devices, the task offloading strategies of multiple devices are coupled [15]. Specifically, multiple devices may simultaneously select the same gateway for task offloading, and the task offloading strategies of each device will be affected by other devices. Last but not least, due to the complex environment of the EIoT, the influence of electromagnetic interference on task offloading optimization should be considered to meet the reliability requirements of data transmission [16].

Matching theory provides a flexible tool to solve the conflicting resource allocation problems. In the matching problem, each agent in one group ranks the agents in the other group according to the utility function to establish a preference list. A stable matching is achieved through the interaction among agents from two sides. Matching theory has been widely applied in task offloading optimization for the IoT. In reference [17], a low-complexity and stable task offloading mechanism based on price matching was proposed to minimize the total network delay. In reference [18], Shan et al. proposed a matching-based two-step approach aiming at minimizing the energy consumption of the IoT devices by optimizing the task offloading decision and transmission power. In reference [19], Huang et al. introduced a task-container matching market to provide on-demand offloading services based on system service capability and resource availability. In reference [20], a matching theory-based task for offloading strategy was proposed, aiming at reducing the total IoT network energy. In reference [21], a solution to minimize the network delay from a contract-matching integration perspective was provided. Nevertheless, these previous works only target a single performance metric as either energy efficiency or delay, while the differentiated service demand guarantee oriented to multiple metrics are ignored.

The joint optimization of multiple metrics in task offloading problems, such as delay and energy efficiency, has been also investigated. Zhang et al. proposed a privacy protection task offloading method based on a deep Q-network (DQN), which can enable users to make optimal offloading decisions to reduce delay and energy consumption while improving privacy levels [22]. Yang et al. proposed a two-stage offloading algorithm, which optimizes computation and communication resource allocation under limited energy and sensitive latency [23]. Chen et al. proposed a two-timescale resource allocation mechanism based on matching theory to minimize power consumption and ensure low-delay performance [24]. In reference [25], a distributed device-to-device offloading system which can guarantee high probability of on-time task completion and low energy consumption was proposed. In reference [26], Ding et al. investigated a decentralized partitioning computation offloading strategy for multiple devices and multiple mobile edge servers with limited resources, where the weighted sum of energy consumption and delay is maximized by optimizing the execution location, CPU frequency, and transmission power. However, the competition conflict of multiple devices in task offloading is not taken into consideration by these works, which cannot achieve stable matching between gateways and devices.

In this work, we propose a many-to-one two-sided matching-based delay and an energy-efficiency-balanced task offloading algorithm for the EIoT to maximize the weighted difference of energy efficiency and delay. First, the task offloading strategies of multiple devices are decoupled through computing resource allocation and the quota settings of the gateways. Then, the formulated problem is transformed into a many-to-one two-sided matching problem between the devices and gateways. Next, the proposed algorithm jointly optimizes the energy efficiency and delay of the task offloading devices, which establishes a stable matching between devices and gateways through continuous iterations. The contributions of this paper are summarized as follows:

- **Dynamic Tradeoff between Energy Efficiency and Delay:** We consider differentiated QoS requirements by establishing the weighted difference between energy efficiency and delay as an optimization objective. The tradeoff between energy efficiency and delay can be dynamically balanced by adjusting the weight according to the requirements of task offloading optimization.
- **Low-Complexity and Stable Task Offloading:** A low-complexity many-to-one two-sided matching-based algorithm is proposed to establish the stable matching between devices and gateways to solve the problem of task offloading conflicts for the EIoT. The preference lists of devices and gateways are modeled as the energy efficiency and total task offloading delay, respectively.
- **Extensive Performance Simulation:** Numerous results demonstrate that the proposed algorithm can achieve superior performance in terms of energy efficiency, delay, and the weighted difference between them compared with existing state-of-the-art algorithms. Moreover, the impact of key parameters such as computing resources and SINR threshold on performance are revealed to provide guidance for practical applications.

The remaining parts of the paper are organized as follows: System model and problem formulation are introduced in Section 2. Section 3 describes the proposed many-to-one two-sided matching-based delay and energy-efficiency-balanced task offloading algorithm for the EIoT. The simulation results are provided in Section 4. Section 5 concludes the paper.

## 2. System Model and Problem Formulation

In this section, we first introduce the task offloading model, including the data transmission model, delay model, and the energy efficiency model. Then, the task offloading optimization problem is formulated.

### 2.1. Task Offloading Model

The considered the task offloading model for the EIoT, which is shown in Figure 1, which consists of multiple gateways and EIoT devices. The gateway establishes wireless connections with devices and provides computing resources. We consider that each device is in the coverage of multiple gateways. Due to the limited computing resources and constrained battery capacity of EIoT devices, the tasks need to be offloaded to the gateway for processing.

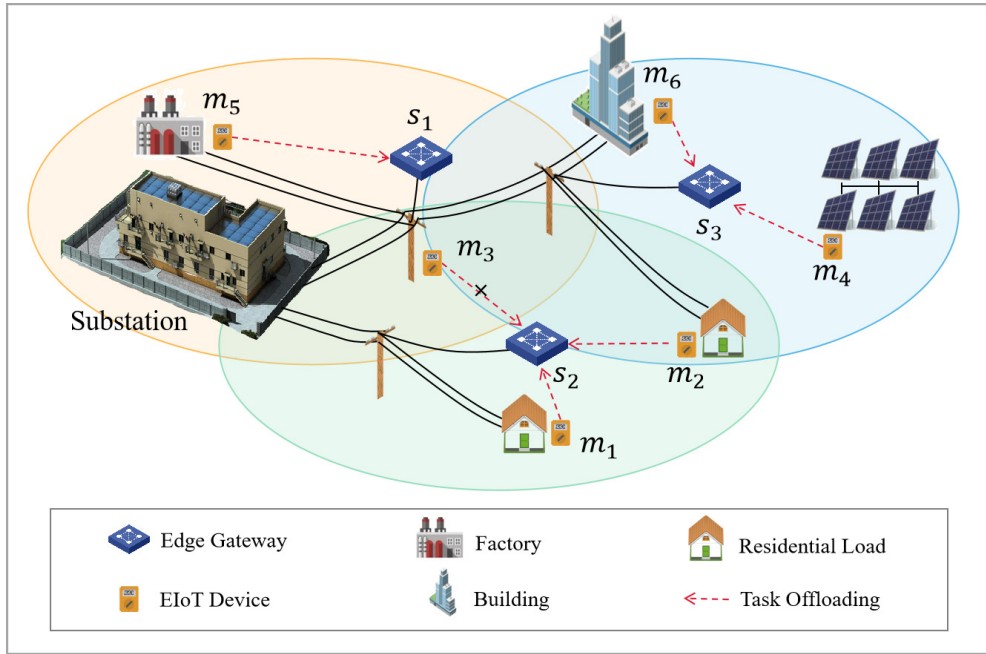

**Figure 1.** Task offloading model for EIoT.

We assume that there are $I$ EIoT devices, the set of which is denoted as $\mathcal{M} = \{m_1, \ldots, m_i, \ldots, m_I\}$. There exist $J$ gateways, the set of which is defined as $\mathcal{S} = \{s_1, \ldots, s_j, \ldots, s_J\}$. We denote the task offloading indicator variable as $x_{i,j}$, where $x_{i,j} = 1$ represents that device $m_i$ selects the gateway $s_j$ for task offloading, and $x_{i,j} = 0$ otherwise. We define the maximum number of EIoT devices that can be simultaneously served as $q_j$. Therefore, $x_{i,j}$ needs to satisfy

$$\sum_{i=1}^{I} x_{i,j} \leq q_j. \tag{1}$$

Figure 1 shows an example of task offloading with six EIoT devices, three gateways, and $q_j = 2$. Task offloading conflict occurs when $m_1$, $m_2$, and $m_3$ simultaneously select $s_2$ for task offloading. Finally, $s_2$ chooses to serve $m_1$ and $m_2$, and $m_3$ remains idle.

### 2.1.1. Data Transmission Model

The EIoT devices can offload tasks to one of the multiple gateways through orthogonal frequency division multiplexing (OFDM)-based orthogonal subchannels (see references [27,28]), and electromagnetic interference is considered. Therefore, the transmission rate between $m_i$ and $s_j$ is given by

$$R_{i,j} = B \log_2(1 + SINR_{i,j}), \tag{2}$$

where $B$ represents the subchannel bandwidth. $SINR_{i,j}$ represents the signal-to-interference-plus-noise ratio (SINR) [29] between $m_i$ and $s_j$, which is given by

$$SINR_{i,j} = \frac{x_{i,j} P^{TX} g_{i,j}}{\sigma^2 + \varepsilon_{i,j}}, \tag{3}$$

where $P^{TX}$ and $\sigma^2$ represent the transmission power and noise power, respectively [30]. Furthermore, $g_{i,j}$ is the channel gain between $m_i$ and $s_j$. $\varepsilon_{i,j}$ represents the electromagnetic interference. Considering the reliability constraints of task offloading, the SINR must satisfy $SINR_{i,j} \geq SINR_{min}$, where $SINR_{min}$ is the SINR threshold.

### 2.1.2. Delay Model

We define $U_i$ as the task data size of $m_i$. The task computing intensity, i.e., the CPU cycles processing one bit by $s_j$, is defined as $f_j$ (cycle/bit). The computing resource (cycle/s) allocated by $s_j$ is defined as $\alpha_j$, which is given by

$$\alpha_j = \frac{\psi_j}{q_j}. \tag{4}$$

Therefore, the transmission delay between $m_i$ and $s_j$, and the computing delay of $s_j$ processing the tasks of $m_i$ are given by

$$Q_{i,j}^{TX} = \frac{U_i}{R_{i,j}}, \tag{5}$$

$$Q_{i,j}^{C} = \frac{f_j U_i}{\alpha_j}. \tag{6}$$

The total task offloading delay is the sum of the transmission delay and computing delay, which is given by

$$Q_{i,j} = Q_{i,j}^{TX} + Q_{i,j}^{C}. \tag{7}$$

### 2.1.3. Energy Efficiency Model

We define $E_{i,j}$ as the task offloading energy consumption of $m_i$, which is given by

$$E_{i,j} = (P^{TX} + P_0)Q_{i,j}^{TX} = (P^{TX} + P_0)\frac{U_i}{R_{i,j}}, \tag{8}$$

where $P_0$ represents the circuit power of the device operation.

We define energy efficiency (bit/(J·Hz)) as the amount of data that can be transmitted per unit of energy per unit of bandwidth [31]. Therefore, the energy efficiency of $m_i$ offloading tasks to $s_j$ is given by

$$\eta_{i,j} = \frac{U_i}{BE_{i,j}} = \frac{U_i}{B(P^{TX} + P_0)\frac{U_i}{R_{i,j}}}$$

$$= \frac{R_{i,j}}{B(P^{TX} + P_0)}. \tag{9}$$

### 2.2. Problem Formulation

In this paper, we address the task offloading problem for the EIoT. The objective is to maximize the weighted difference between energy efficiency and delay $\Psi(x_{i,j})$ under the constraints of task offloading and SINR. As such, $\Psi(x_{i,j})$ is given by

$$\Psi(x_{i,j}) = \sum_{i=1}^{I}\sum_{j=1}^{J} x_{i,j}(V\eta_{i,j} - Q_{i,j}), \tag{10}$$

where $V$ is the weight to balance the order of magnitude between energy efficiency and delay. The task offloading optimization problem is modeled as

$$\mathbf{P1}: \max_{\{x_{i,j}\}} \Psi(x_{i,j}),$$

$$\text{s.t. } C_1: \sum_{i=1}^{I} x_{i,j} \leq q_j, \ s_j \in \mathcal{S}, \ x_{i,j} \in \{0,1\},$$

$$C_2: \sum_{j=1}^{J} x_{i,j} = 1, \ m_i \in \mathcal{M}, \ x_{i,j} \in \{0,1\},$$

$$C_3: SINR_{i,j} \geq SINR_{min}, \ m_i \in \mathcal{M}, \ s_j \in \mathcal{S}. \tag{11}$$

$C_1$ and $C_2$ are the task offloading constraints, which indicate that each gateway can be selected by $q_i$ devices at most, and each device can select only one gateway. $C_3$ is the SINR constraint.

## 3. Delay and Energy-Efficiency-Balanced Task Offloading for EIoT Based on Two-Sided Matching

In this section, we elaborate the problem transformation and the proposed delay and energy-efficiency-balanced task offloading algorithm for the EIoT based on two-sided matching.

### 3.1. Problem Transformation

To solve **P1**, it is feasible to traverse all possible results due to the high computational complexity [32,33]. Therefore, we convert **P1** into a many-to-one two-sided matching problem between devices and gateways.

**Theorem 1.** *Matching: Matching $\phi$ is defined as the mapping relationship of a set $\mathcal{M} \cup \mathcal{S}$, which is denoted as $\phi: \mathcal{M} \cup \mathcal{S} \to \mathcal{S} \cup \mathcal{M}$. When $\phi(m_i) = s_j$ and $\phi(s_j) = m_i$, it means that $m_i$ and $s_j$ establish a matching relationship, which can be expressed as*

$$x_{i,j} = \begin{cases} 1, \phi(m_i) = s_j \text{ and } \phi(s_j) = m_i, \\ 0, \text{otherwise.} \end{cases} \tag{12}$$

In many-to-one two-side matching, each device traverses all gateways to obtain the preference values toward different gateways and each gateway traverses all devices to obtain the preference values toward different devices. The preference value of $m_i$ for $s_j$ and $\omega_{i,j}$, is defined as the weighted energy efficiency of $m_i$ offloading tasks to $s_j$. The preference value of $s_j$ for $m_i$ and $\gamma_{j,i}$, is defined as the total delay of $m_i$ offloading tasks to $s_j$. Furthermore, $\omega_{i,j}$ and $\gamma_{j,i}$ are expressed as

$$\omega_{i,j} = V \cdot \eta_{i,j}, \tag{13}$$

$$\gamma_{j,i} = -Q_{i,j}. \tag{14}$$

For each $m_i$ and $m_i \in \mathcal{M}$, $s_j \succ_{m_i} s_{j'}$ denotes that $m_i$ prefers $s_j$ to $s_{j'}$, which is given by

$$s_j \succ_{m_i} s_{j'} \Leftrightarrow \omega_{i,j}|_{\phi(m_i)=s_j} > \omega_{i,j'}|_{\phi(m_i)=s_{j'}}, \tag{15}$$

where $\Leftrightarrow$ is a binary preference relationship.

Meanwhile, for each $s_j$ and $s_j \in \mathcal{S}$, $m_i \succ_{s_j} m_{i'}$ denotes that $s_j$ prefers $m_i$ to $m_{i'}$, which is given by

$$m_i \succ_{s_j} m_{i'} \Leftrightarrow \gamma_{j,i}|_{\phi(s_j)=m_i} > \gamma_{j,i'}|_{\phi(s_j)=m_{i'}}. \tag{16}$$

After obtaining $\omega_{i,j}|_{\phi(m_i)=s_j}, s_j \in \mathcal{S}$ and $\gamma_{j,i}|_{\phi(s_j)=m_i}, m_i \in \mathcal{M}$, the preference list of devices is obtained by sorting $\omega_{i,j}|_{\phi(m_i)=s_j}, s_j \in \mathcal{S}$ in descending order, which is denoted as

$$\mathcal{F}_i = \{\cdots, s_j, s_{j'} \cdots\}, s_j \succ_{m_i} s_{j'}. \tag{17}$$

The preference list of gateways is obtained by sorting $\gamma_{j,i}|_{\phi(s_j)=m_i}$ in descending order, which is denoted as

$$\mathcal{F}_j = \{\cdots, m_i, m_{i'} \cdots\}, m_i \succ_{s_j} m_{i'}. \tag{18}$$

Based on $\mathcal{F}_i$ and $\mathcal{F}_j$, **P1** can be solved by the proposed delay and energy-efficiency-balanced task offloading algorithm.

*3.2. Many-to-One Two-Sided Matching-Based Delay and Energy-Efficiency-Balanced Task Offloading Algorithm*

The implementation process of the proposed delay and energy-efficiency-balanced task offloading algorithm is shown in Algorithm 1, which consists of three steps.

***Step 1:*** Initialize the sets of task offloading strategies, unmatched devices, and unmatched gateways as $\phi = \varnothing$, $\Omega = \mathcal{M}$ and $\Gamma = \mathcal{S}$;

***Step 2:*** Each $m_i$ and $m_i \in \mathcal{M}$, and each $s_j$ and $s_j \in \mathcal{S}$ calculate the preference values based on (13) and (14), and establish the preference lists $\mathcal{F}_i$ and $\mathcal{F}_j$ based on (17) and (18);

***Step 3:*** First, each device in $\Omega$ proposes to its most preferred gateway based on $\mathcal{F}_i$, i.e., the top gateway in its preference list.

Afterwards, each $s_j$ and $s_j \in \mathcal{S}$ calculates the total number of temporary matches and new proposals. If the total number of temporary matches and new proposals is less than quota $q_i$, $s_j$ establishes temporary matches with the devices which propose to it. The

matched devices are temporarily removed from $\Omega$. Otherwise, $s_j$ establishes temporary matches with only the top $q_j$ devices in its preference list $\mathcal{F}_j$, which propose to it. Then, the matched devices are removed from $\Omega$. The unmatched devices are added into $\Omega$ and remove $s_j$ from $\mathcal{F}_i$. If the total number of matches for $s_j$ is equal to $q_j$, remove $s_j$ from $\Gamma$.

Iterative matching ends when each device establishes a match with a gateway or its preference list $\mathcal{F}_i = \varnothing$. Based on the matching results, each device offloads the tasks to the corresponding gateway for data processing.

---

**Algorithm 1** Delay and Energy-Efficiency-balanced Task Offloading Algorithm

---

1: **Input**: $\mathcal{M}$, $\mathcal{S}$.
2: **Output**: $\phi$.
3: **Step 1: Initialization**
4:     Initialize $\phi = \varnothing$, $\Omega = \mathcal{M}$, and $\Gamma = \mathcal{S}$.
5: **Step 2: Preference List Construction**
6:     Each $m_i$ and $s_j$ calculate the preference values based on (13) and (14), and establish the preference lists $\mathcal{F}_i$ and $\mathcal{F}_j$ based on (17) and (18).
7: **Step 3: Iterative Matching**
8: **While** $\Omega \neq \varnothing$ and $\mathcal{F}_i \neq \varnothing$ **do**
9:     **For** $m_i \in \Omega$ **do**
10:         $m_i$ proposes to its most preferred gateway based on $\mathcal{F}_i$.
11:     **End for**
12:     **For** $s_j \in \Gamma$ **do**
13:         **If** the total number of temporary matches and new proposals, e.g., $m_i$, for $s_j$ is less than quota $q_j$ **then**
14:             Match $s_j$ with the devices which propose to it temporarily and remove the matched devices from $\Omega$.
15:         **else**
16:             Temporarily match $s_j$ with its most preferred $q_j$ devices proposing to $s_j$ based on $\mathcal{F}_j$. Remove matched devices from $\Omega$ and add unmatched devices into $\Omega$.
17:             Unmatched devices update $\mathcal{F}_i = \mathcal{F}_i \setminus s_j$.
18:         **End if**
19:         **If** the total number of matches for $s_j$ is equal to $q_i$ **then**
20:             $\Gamma = \Gamma \setminus s_j$.
21:         **End if**
22:     **End for**
23: **End while**

---

### 3.3. Complexity Analysis

The complexity of the proposed algorithm is $O(I + J + I\log(J) + J\log(J)) + O(I)$. Based on the analysis, the complexity of the proposed algorithm has a negligible impact on the delay performance during task offloading and is also applicable when the number of devices is large.

## 4. Simulation Results

In this section, we first introduce the simulation parameter settings. Afterwards, extensive simulation results are elaborated to verify the effectiveness of the proposed algorithm. The simulation is implemented on MATLAB R2021.

### 4.1. Simulation Parameter Settings

The proposed algorithm is compared with two existing task offloading algorithms, i.e., an energy-efficiency-first (EEF) task offloading algorithm [34], which aims to maximize the the energy efficiency without considering the time delay. The second is a time-delay-first (TDF) task offloading algorithm [13], which aims to minimize the time delay without considering the energy efficiency. The simulation results are shown from Figures 2–9. Furthermore, to verify the performance of the proposed algorithm, we compare the proposed

algorithm with the task offloading algorithm based on $\varepsilon$-greedy, which jointly optimizes the weighted difference between energy efficiency and delay [35]. Through 500 experiments, the simulation comparison is performed and shown from Figures 10–12.

The channel gain is influenced by the small-scale fading, i.e., $g_{i,j} = (127 + 30 \times \log d_{i,j})$, where $d_{i,j}$ is the distance between $m_i$ and $s_j$. Other parameters are summarized in Table 1 [34,36].

**Table 1.** Simulation parameters.

| Parameters | Value |
|:---:|:---:|
| $I$ | $[60, 100]$ |
| $q_j$ | $[10, 20]$ |
| $U_i$ | $[1.5, 2]$ Mbits |
| $\alpha_j$ | $[6 \times 10^9, 5 \times 10^{10}]$ cycle/s |
| $f_j$ | $[1 \times 10^3, 3.5 \times 10^3]$ cycle/bit |
| $P_0, P_{TX}$ | 0.1 W, 0.4 W |
| $V$ | 25 |
| $\sigma^2$ | $-174$ dBm |
| $SINR_{min}$ | 16 dB |
| $B$ | 0.2 MHz |

*4.2. Simulation Results and Analysis*

Figure 2 shows the weighted difference between energy efficiency and delay versus the number of devices. The EEF and TDF only consider a single optimization objective. It is difficult to ensure that the delay and energy efficiency are optimized at the same time. Therefore, the proposed algorithm sets the weight $V$ to unify the device energy efficiency and delay magnitude, and builds the weighted difference between energy efficiency and delay based on the weight $V$ to realize the joint optimization of energy efficiency and delay. The dynamic compromise between energy efficiency and transmission delay improves the overall performance of the proposed algorithm. The simulation results show that as the number of devices increases, the weighted difference between energy efficiency and delay under different algorithms decrease. The performance of the proposed algorithm is always better than that of EEF and TDF. When the number of devices is 120, compared with EEF and TDF, the performances of the proposed algorithm are increased by 29.01% and 45.65%, respectively.

Figure 3 shows the weighted difference between energy efficiency and delay versus the number of gateways. The simulation results show that the weighted difference improves with the number of gateways. The proposed algorithm always has the best performance among the three algorithms. Compared with EEF and TDF, the proposed algorithm can enhance the weighted difference between energy efficiency and delay by 11.57% and 22.25%, respectively, when the number of gateways is up to 16.

Figure 4 shows the computing and transmission delay versus $q_j$, and Figure 5 shows the total delay, energy efficiency, and weighted difference versus $q_j$. The simulation results show that the computing delay, energy efficiency, and weighted difference are positively proportional to $q_j$, while the transmission delay decreases with $q_j$. The reason is that as $q_j$ becomes larger, less computing resources are allocated to each device, thereby increasing computing delay. Meanwhile, the device has a larger probability to offload tasks to a closer gateway, so as to reduce the transmission delay. When gateways can accept more devices, the device will access the gateway with higher energy efficiency, which increases the performance of energy efficiency. Therefore, it is necessary to select the appropriate values of $q_i$ to realize the dynamic tradeoff among computing delay, transmission delay, and energy efficiency.

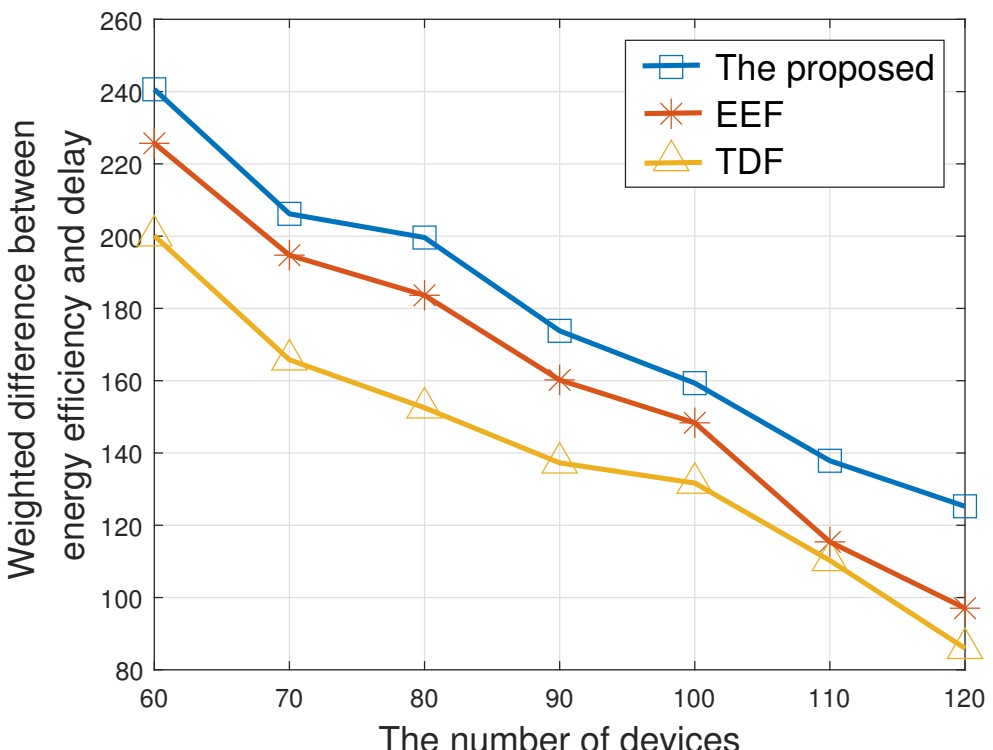

**Figure 2.** Weighted difference between energy efficiency and delay versus the number of devices.

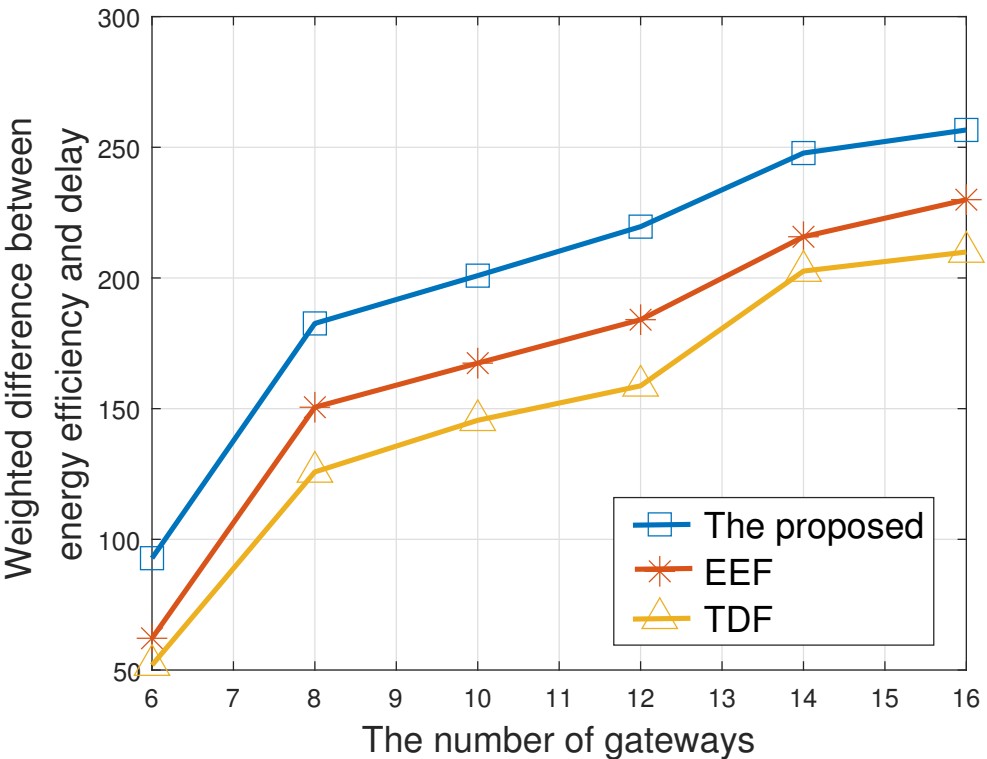

**Figure 3.** Weighted difference between energy efficiency and delay versus the number of gateways.

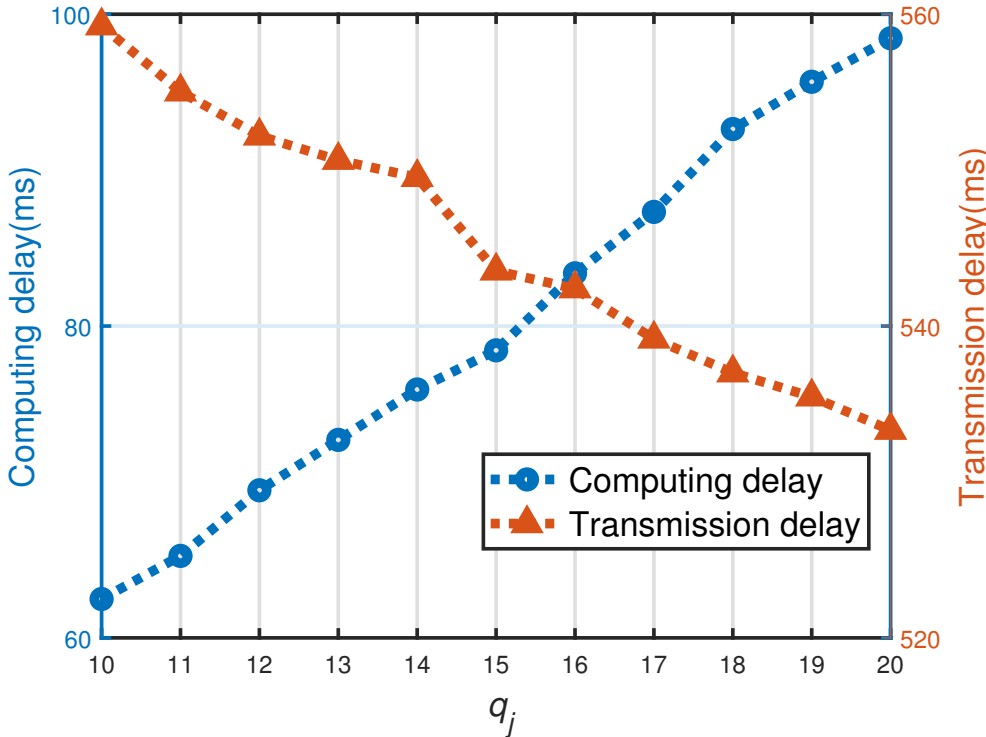

**Figure 4.** Computing delay and transmission delay versus $q_j$.

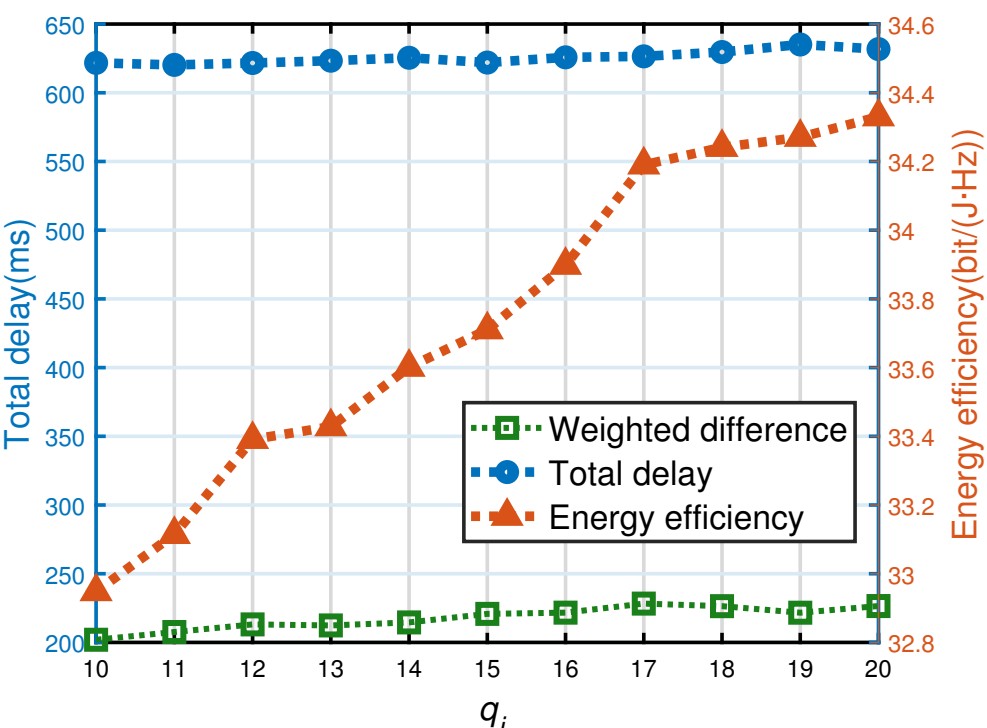

**Figure 5.** Total delay, energy efficiency, and weighted difference versus $q_j$.

Figure 6 shows the total delay and energy efficiency versus computing resources. The simulation results demonstrate that when computing resources of gateways are increased from $4 \times 10^{10}$ to $8 \times 10^{10}$, the energy efficiency keeps increasing trend while the total delay keeps decreasing trend. The reason is that larger computing resources reduce the computing delay for devices during the task offloading process, thereby reducing the total delay. Moreover, sufficient computing resources encourage devices to select closer gateways for task offloading, which leads to higher energy efficiency.

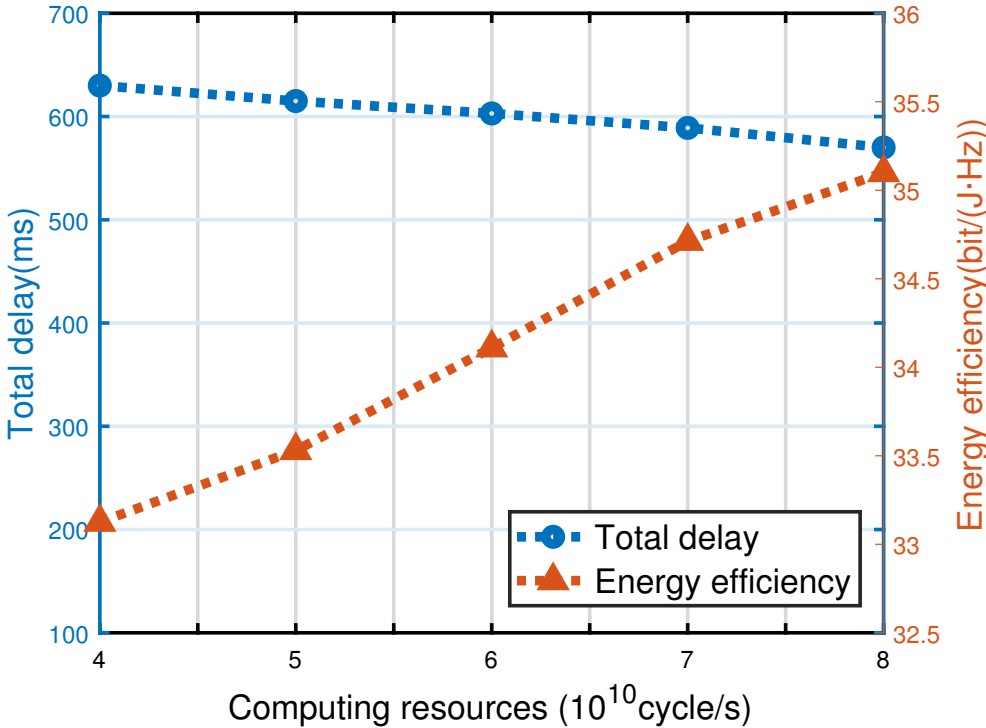

**Figure 6.** Total delay and energy efficiency versus computing resources.

Figure 7 shows shows total delay and energy efficiency versus $SINR_{min}$. The simulation results demonstrate that the energy efficiency increases with $SINR_{min}$, while the total delay almost remains unchanged. The reason is that a larger $SINR_{min}$ results in a tighter SINR constraint, which cannot be satisfied by the remote gateways with a poor SINR. Devices are enforced to choose closer gateways for task offloading. Therefore, the energy efficiency is increased and transmission delay is decreased due to the reduced transmission distance. However, the computing resources of remote gateways cannot be utilized, which increases the computing delay. Therefore, the total delay almost remains unchanged.

Figure 8 shows the weighted difference between energy efficiency and delay versus computing resources. The simulation results demonstrate that the weighted difference between the energy efficiency and delay of all three algorithms increase with computing resources, and the proposed algorithm increases fastest. Based on Figure 9, with the increase of computing resources, the computing delay is reduced and the energy efficiency is increased, resulting in an increase in the weighted difference between energy efficiency and delay. EEF has the smallest increasing trend. The reason is that EEF only optimizes energy efficiency and the computing delay will not affect the gateway selection of devices.

Figure 9 shows the weighted difference between energy efficiency and delay versus $SINR_{min}$. The simulation results demonstrate that the weighted difference between the energy efficiency and delay of the proposed algorithm and the TDF increases with $SINR_{min}$, while that of the EEF remains almost unchanged. The reason is that the EEF always selects the gateway with a better SINR for task offloading according to the energy efficiency optimization objective, and the increase of $SINR_{min}$ will hardly change the gateway selection.

Since the proposed algorithm takes into account both energy efficiency and delay for the task offloading optimization, it performs better than the TDF.

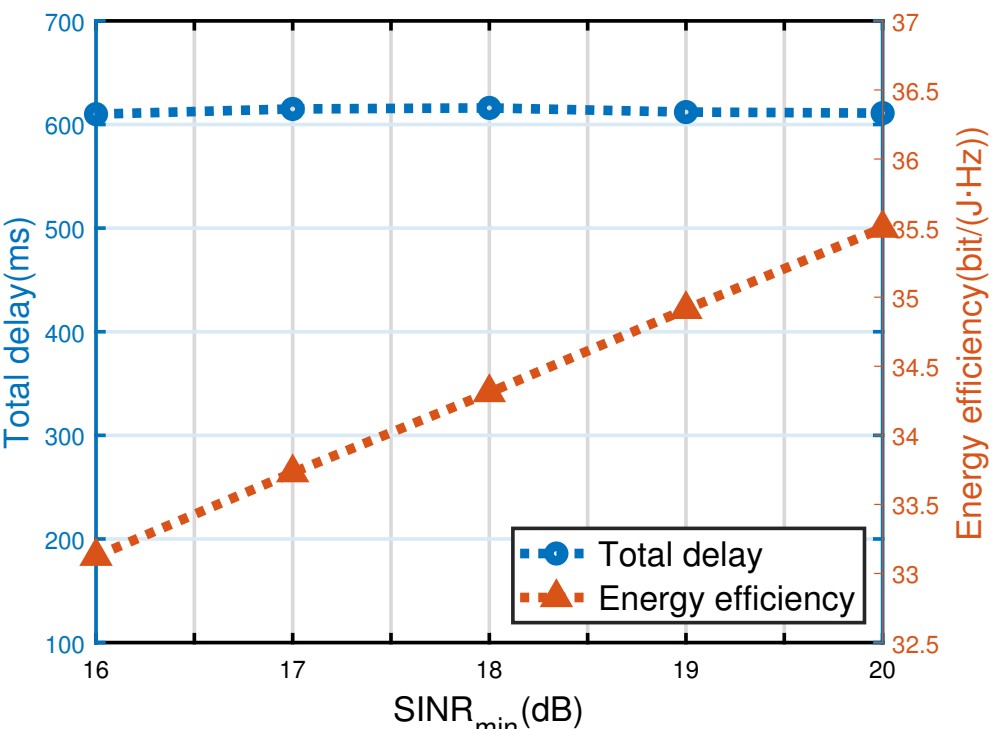

**Figure 7.** Total delay and energy efficiency versus $SINR_{min}$.

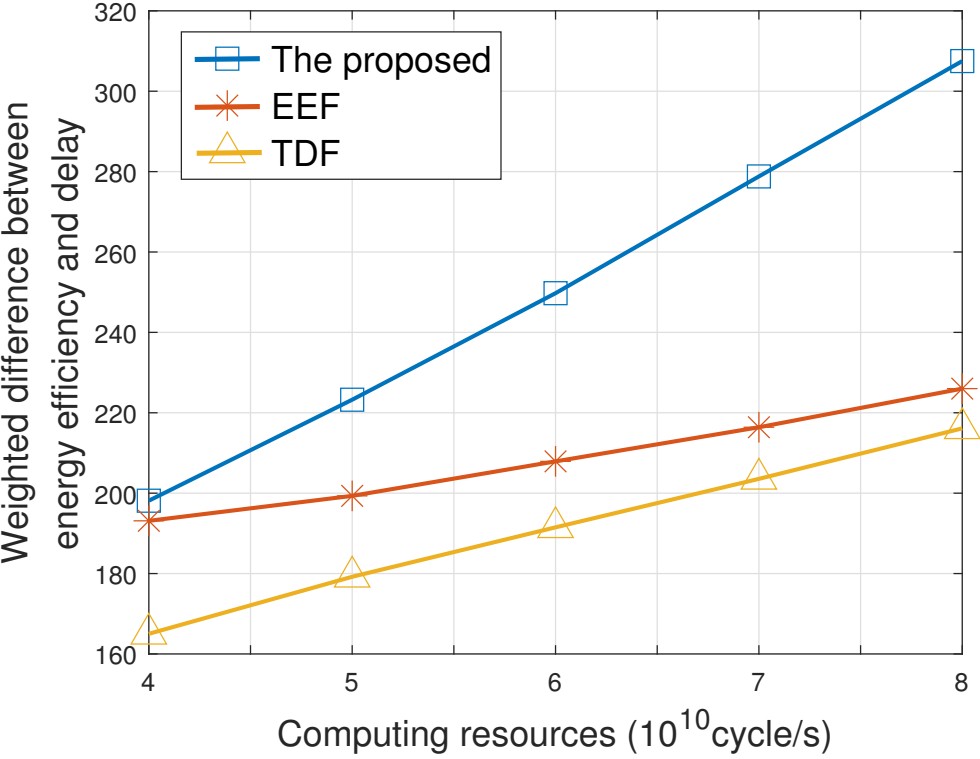

**Figure 8.** Weighted difference between energy efficiency and delay versus computing resources.

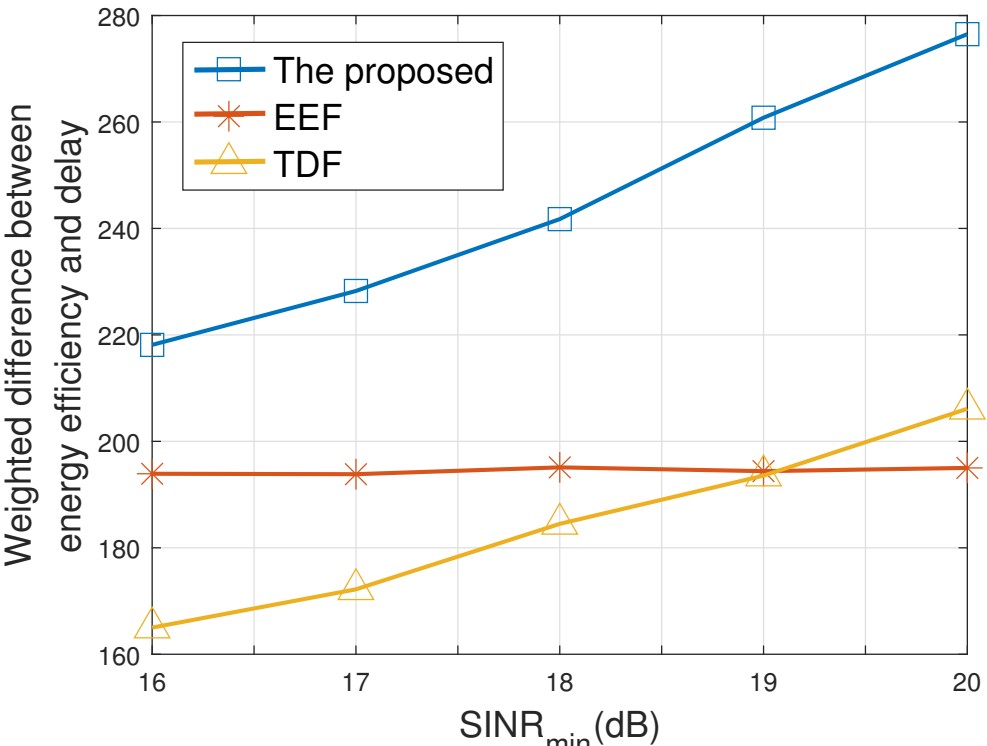

**Figure 9.** Weighted difference between energy efficiency and delay versus $SINR_{min}$.

Figure 10 shows the weighted difference between energy efficiency and delay versus simulation numbers. When the number of tests increases, the weighted difference of the $\varepsilon$-greedy algorithm increases, but the performance of the proposed algorithm is much better than the greedy algorithm. When the number of tests is 500, the proposed algorithm has a 73.43% increase in weighted difference between energy efficiency and delay compared to the $\varepsilon$-greedy algorithm. The reason is that the $\varepsilon$-greedy algorithm needs to explore non-optimal options to obtain task offloading strategies and ignores the coupling among different devices. The proposed algorithm can reduce the complexity of the coupling problem and realize the precise matching between devices and gateways to achieve the joint optimization of the energy efficiency and delay of all devices.

Figure 11 shows the energy efficiency versus simulation numbers. Figure 12 shows the total delay versus simulation numbers. After many simulations, the performances of the $\varepsilon$-greedy algorithm in terms of energy efficiency and total delay have been improved but are still lower than the proposed algorithm. This is because the $\varepsilon$-greedy algorithm has difficulty in solving the coupling among different devices.

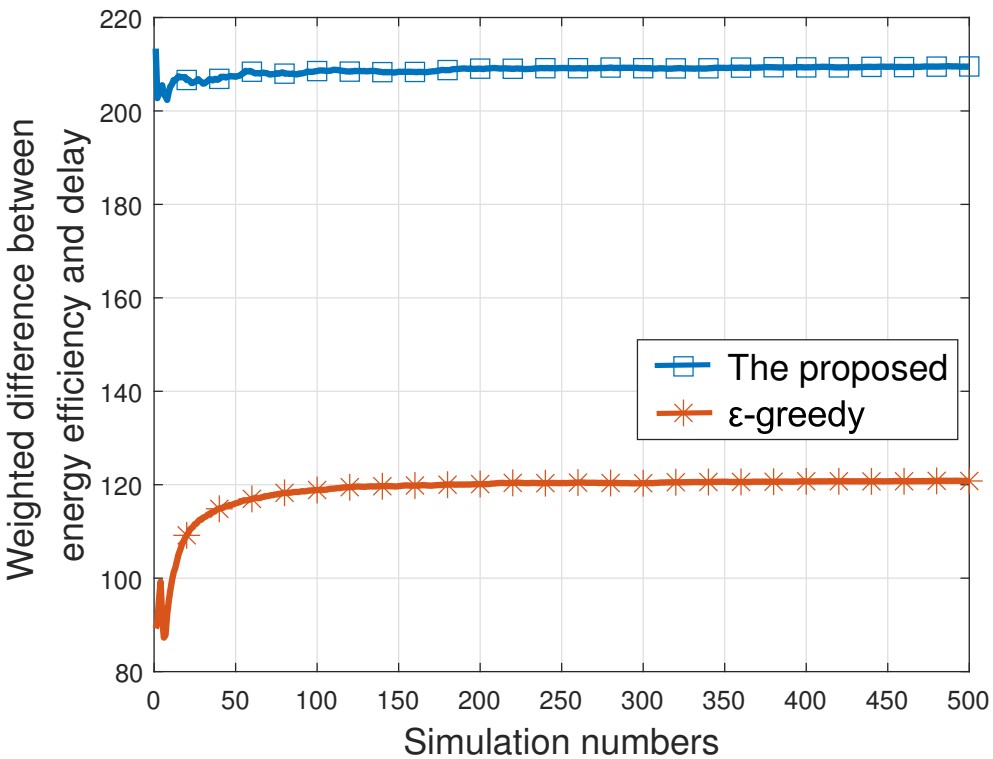

**Figure 10.** Weighted difference between energy efficiency and delay versus simulation numbers.

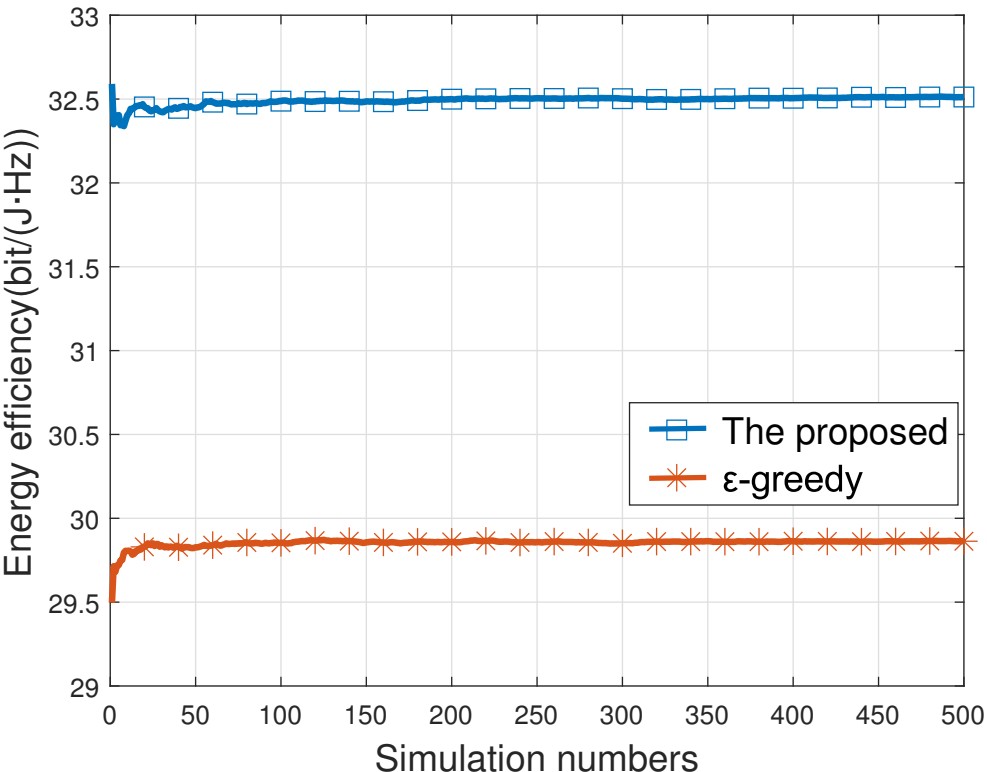

**Figure 11.** Energy efficiency simulation numbers.

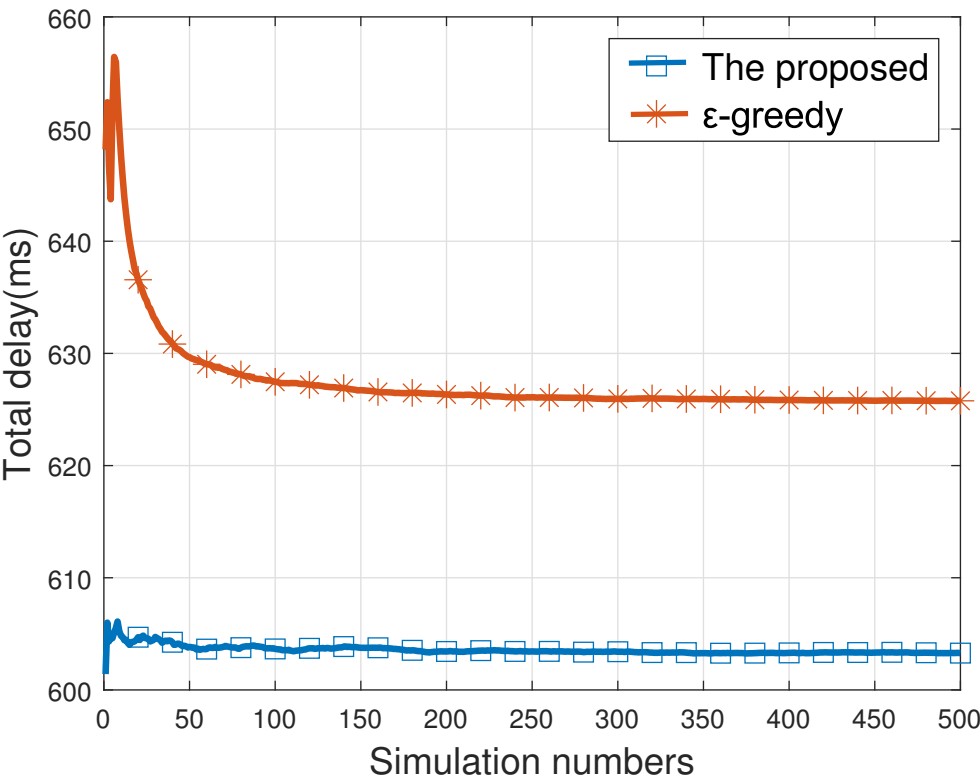

**Figure 12.** Total delay versus simulation numbers.

## 5. Conclusions

In this paper, we addressed the task offloading problem for the EIoT, and proposed a delay and energy-efficiency-balanced task offloading algorithm. We first modeled the optimization objective of the task offloading problem as the weighted difference between delay and energy efficiency. Then, the task offloading problem was transformed into a many-to-one two-sided matching problem, where the preference values of the devices and gateways in regard to each other were calculated based on energy efficiency and delay, respectively. A stable matching between devices and servers was established through a continuous iteration. Therefore, the proposed algorithm achieved a dynamic tradeoff between energy efficiency and delay as well as low-complexity and stable task offloading. The simulation results showed that compared with EEF and TDF task offloading algorithms, the proposed algorithm could improve the weighted difference between energy efficiency and delay by 29.01% and 45.65% when the number of devices is 120, and by 11.57% and 22.25% when the number of gateways is 16, respectively. In addition, compared with an $\epsilon$-greedy task offloading algorithm, the proposed algorithm had a 73.43% increase in weighted difference between energy efficiency and delay when the simulation number is 500. Some perspectives related to future work are outlined.

**Task Offloading Optimization under Uncertain Information:** Considering the complex and dynamic environment of the EIoT, collecting the global state information, such as the computing resource of gateways, the channel state information, and the task offloading decision of other devices, incurs prohibitive signaling overheads. Therefore, the key information for task offloading optimization is uncertain. In the future, we will consider exploring advanced artificial intelligence (AI) with powerful environment-learning capabilities to deal with task offloading optimization under uncertain information conditions.

**Joint Optimization of Task Offloading and Computing Resource Allocation:** We only consider task offloading optimization and assume that gateways allocate computing resources equally to devices in this paper. However, the equal allocation manner cannot ensure the efficient utilization of computing resources. Specifically, computing resources allocated to the devices with large amounts of offloaded data may be insufficient, resulting

in large computing delays, while computing resources allocated to other devices may be idle. In the future, we will explore the joint optimization of task offloading and computing resource allocation to further improve the delay performance of devices and the resource utilization of gateways.

**Author Contributions:** Conceptualization, Y.W., H.Y., and J.W.; methodology, Y.W., X.C., and J.L.; software, Y.W. and S.Z.; validation, Y.W., H.Y., J.W., X.C., J.L., S.Z., and B.H.; formal analysis, H.Y., J.W., and B.H.; investigation, Y.W., X.C., and J.L.; data analysis, Y.W., H.Y., S.Z., and B.H.; writing—original draft preparation, Y.W., H.Y., and S.Z.; writing—review and editing, Y.W., H.Y., J.W., X.C., J.L., S.Z., and B.H. All authors have read and agreed to the published version of the manuscript.

**Funding:** This work was supported by the Science and Technology Project of the State Grid Corporation of China under Grant Number 5204XA20004K.

**Institutional Review Board Statement:** Not applicable.

**Informed Consent Statement:** Not applicable.

**Data Availability Statement:** Not applicable.

**Conflicts of Interest:** The authors declare no conflict of interest.

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
