# Peer review of "Delay and Energy-Efficiency-Balanced Task Offloading for Electric Internet of Things"

_electronics, doi:10.3390/electronics11060839_

Round 1
Reviewer 1 Report
The topic of the manuscript fits well the scope of the journal where the authors have written a good organized research article about this topic. The introduction in Section 1, system model and problem formulation in Section 2, and delay and energy efficiency-balanced task offloading for EIoT based on two-sided matching in Section 3 are described mostly correctly. However, authors should address some issues before the manuscript can be considered for acceptance at Electronics-MDPI, Here they are:
- The abstract requires more improvements by showing the novelty and most deliverables of this work.
- The simulation results in Section 4 are simplistic and not well discussed and a lot of work has gone into writing this manuscript and conceptualizing the model.
- The conclusion in Section 5 is of little value, therefore, conclusion should state clear paragraphs or bullet points related to accomplishments of objectives and novelty of the work that is supported by some numeric results. Also, some perspectives related to future work can be added in this section.
Author Response
Thank you very much for your in-detail reading of the manuscript. We would like to express our sincere appreciation and thankfulness for your insightful and valuable comments. We have made great efforts to answer your questions and further improve the quality of the manuscript by following your recommendations. We hope that the revised manuscript would be suitable to meet the publication standard of Electronics. The following is a point by point response to your numbered comments. Again, we would like to express our sincere gratitude for your time and consideration. Please see the attachment for detail response.

Reviewer 2 Report
This paper shows a practical case study. The subject of the paper is very interesting and worth of study. The analysis of the studied topics in this paper is clear and there is a good link between the results and the conclusion of the study. However, there are some comments:
- The conclusion can be improved to include more details about the study results.
- The high lights are missing in the paper
- It would be easier for the reader, if you could describe the used simulation program in more detail.
Author Response

(The authors gave the same response as above.)

Round 2
Reviewer 1 Report
The manuscript has been reviewed properly and it is suitable for publishing in Electronics-MDPI.